# Relationship between Bullying Victimization and Quality of Life in Adolescents with Attention-Deficit/Hyperactivity Disorder (ADHD) in Taiwan: Mediation of the Effects of Emotional Problems and ADHD and Oppositional Defiant Symptoms

**DOI:** 10.3390/ijerph18189470

**Published:** 2021-09-08

**Authors:** Chien-Wen Lin, Kun-Hua Lee, Ray C. Hsiao, Wen-Jiun Chou, Cheng-Fang Yen

**Affiliations:** 1Department of Psychiatry, Kaohsiung Medical University Hospital, Kaohsiung 80708, Taiwan; u9901043@cmu.edu.tw; 2Department of Educational Psychology and Counseling, National Tsing Hua University, Hsinchu 300044, Taiwan; kunhualee@mx.nthu.edu.tw; 3Department of Psychiatry and Behavioral Sciences, University of Washington School of Medicine, Seattle, WA 98195, USA; rhsiao@u.washington.edu; 4Department of Psychiatry, Children’s Hospital and Regional Medical Center, Seattle, WA 98105, USA; 5College of Medicine, Chang Gung University, Taoyuan 33302, Taiwan; 6Department of Child and Adolescent Psychiatry, Chang Gung Memorial Hospital, Kaohsiung Medical Center, Kaohsiung 83301, Taiwan; 7Department of Psychiatry, School of Medicine, College of Medicine, Kaohsiung Medical University, Kaohsiung 80708, Taiwan; 8College of Professional Studies, National Pingtung University of Science and Technology, Pingtung 91201, Taiwan

**Keywords:** attention-deficit/hyperactivity disorder, bullying victimization, children, emotional problems, oppositional defiant disorder, psychological well-being, quality of life

## Abstract

This cross-sectional study investigated the mediating effects of emotional problems including depression, anxiety, attention-deficit/hyperactivity disorder (ADHD), and oppositional defiant disorder (ODD) symptoms on the association between bullying victimization and quality of life (QoL) among adolescents with ADHD in Taiwan. A total of 171 adolescents diagnosed as having ADHD participated in this study. Adolescents completed the School Bullying Experience Questionnaire, the Taiwanese Quality of Life Questionnaire for Adolescents, the Taiwanese version of the Children’s Depression Inventory and the Multidimensional Anxiety Scale for Children. Caregivers completed the Chinese version of the Swanson, Nolan, and Pelham Version IV Scale. Structural equation modeling (SEM) was used to examine the relationships among the variables. The results of SEM revealed that bullying victimization indirectly correlated with QoL through the mediation of emotional problems in adolescents with ADHD, whereas ADHD and ODD symptoms did not mediate the association between bullying victimization and QoL. Bullying victimization should be actively prevented and intervened on to ensure better QoL in adolescents with ADHD. Moreover, emotional problems should be alleviated among adolescents with ADHD with bullying victimization experience to maintain their QoL.

## 1. Introduction

### 1.1. Bullying Victimization and Its Association with Quality of Life

Bullying is an aggressive behavior repeated over a time period [1,2]. Bullying behavior can be physical acts (hitting, pushing, and kicking), verbal utterances (name calling, provoking, making threats, and spreading rumors), or other behaviors (making faces or social exclusion) [3]. Bullying takes place within relatively small and stable settings (like classes), which are characterized by the presence of the same people (e.g., children) [4]. Bullying victimization is a distressing experience and considerably affects the lives of children and adolescents [5]. The effects of bullying on emotional health may persist over time. For example, children who were bullied repeatedly through middle adolescence were found to have lower self-esteem and more depressive symptoms after they grew up [6]. Compromised quality of life (QoL) is also an adverse outcome of bullying victimization [7,8,9,10,11,12]. QoL is defined as individuals’ perceptions of their position in life in the context of the culture and value systems in which they live, and in relation to their goals, expectations, standards, and concerns [13]. Given that QoL is an incorporating concept of the persons’ physical health, psychological state, level of independence, social relations, personal beliefs, and relationship to salient features of the environment [13], it is important to understand how bullying victimization compromises the QoL of adolescents.

Attention-deficit/hyperactivity disorder (ADHD) is the most common neurodevelopmental disorder with a prevalence around 5.9–7.2% [14,15]. ADHD is associated with adverse outcomes, including poor academic performance, mental and substance use disorders, criminality, and unemployment [16]. Children and adolescents with ADHD are a high-risk group for bullying victimization. A cross-sectional community study in Sweden found that the diagnosis of ADHD was significantly associated with bullying victimization [17]. Cross-sectional community studies also showed that children [18] and adolescents [19] who report more severe ADHD symptoms are more likely to be bullying victims. A follow-up community study in 10-year-old Korean children found that ADHD symptoms can predict bullying victimization 2 years later [20]. It has been proposed that difficulties in controlling impulsivity and demonstrating appropriate behaviors in social situations increase the vulnerability to bullying victimization in children and adolescents with ADHD [17,21,22,23].

Few studies have examined the association between bullying victimization and QoL in children and adolescents with ADHD. A previous study in Turkey reported no significant difference in the level of QoL between children with ADHD with and without bullying victimization [24]; however, the small sample size and age distribution (mean age = 8.6 ± 1.1 years) limited the generalization of the study results to the population of adolescents with ADHD. Moreover, no study has examined factors that mediate the association between bullying victimization and poor QoL in adolescents with ADHD. Understanding the mediators can help clinicians and public health professionals develop appropriate intervention strategies for improving QoL in adolescents with ADHD experiencing bullying victimization.

### 1.2. Mediating Role of Emotional Problems

The mediating effect of emotional problems on the association between bullying victimization and QoL in adolescents with ADHD has not been examined. Regarding the association between bullying victimization and emotional problems, several cross-sectional studies have determined a significant link between bullying victimization and increased emotional problems among adolescents, including depression [25,26,27,28,29] and anxiety [30,31]. Longitudinal studies have also demonstrated that bullying victimization can lead to future emotional problems [32,33,34,35,36,37,38,39]. A study in the United States of America reported that both relational and physical bullying victimization increased the risk of anxiety in both male and female adolescents with ADHD, whereas relational bullying victimization increased the risk of depression only in male adolescents with ADHD [40].

Regarding the association between emotional problems and QoL, research has reported that emotional problems may reduce the QoL of individuals with ADHD. A longitudinal study indicated that compared with children with ADHD alone, children with ADHD and internalizing comorbidities had poorer family and peer QoL [41]. Another follow-up study demonstrated that co-occurring anxiety and depressive symptoms mediated the effect of childhood ADHD symptoms on QoL in adulthood [42]. The results of previous studies have indicated that bullying victimization, emotional problems, and decreased QoL may interact with each other in individuals with ADHD. However, the mediating effect of emotional problems on the effect of bullying victimization on QoL in adolescents with ADHD warrants further study.

### 1.3. Mediating Roles of ADHD and Oppositional Defiant Disorder Symptoms

The mediating effect of ADHD and oppositional defiant disorder (ODD) symptoms on the association between bullying victimization and QoL in adolescents with ADHD has not been examined. Regarding the association between bullying victimization and ADHD and ODD symptoms, cross-sectional [43] and longitudinal studies [33] have reported a significant association between bullying victimization and hyperactivity and conduct problems. The significant relationship between ADHD symptoms and bullying victimization has been well established [12,44]. For example, in a previous study in Spain, peer victimization was strongly associated with symptoms such as inattention and executive dysfunction among children and adolescents with ADHD [44].

Regarding the association between ADHD and ODD symptoms and QoL, studies indicated that the severity of ADHD symptoms was negatively associated with the QoL of children and adolescents with ADHD [45,46]. ODD is comorbid in more than half of individuals with ADHD [47]. Comorbid symptoms of ODD in individuals with ADHD can have a significant impact on the course and prognosis of the disorder, and may lead to different treatment responses to both behavioral and pharmacologic treatments. Therefore, ODD symptoms are a key consideration in ADHD management [47]; the comorbidity of ODD exerted an additionally deleterious effect on the QoL of individuals with ADHD [48]. Whether ADHD and ODD symptoms mediate the effect of bullying victimization on the QoL of adolescents with ADHD should be investigated. Moreover, the severity of ADHD symptoms was positively associated with the severity of emotional problems such as depressive and anxiety symptoms among children [49,50,51] and young adults with ADHD [52,53]. Thus, whether emotional problems mediate the association between ADHD and ODD symptoms and QoL in adolescents with ADHD should be examined.

### 1.4. Aims of This Study

This cross-sectional study investigated the mediating effects of emotional problems (including depression and anxiety) and ADHD and ODD symptoms on the association between bullying victimization and QoL, in adolescents with ADHD diagnosed based on the criteria of the Diagnostic and Statistical Manual of Mental Disorders, Fourth Edition, Text Revision (DSM-IV-TR) [54]. Figure 1 presents the hypothesized model. We hypothesized that after controlling the effects of gender and age: (1) bullying victimization would be negatively associated with QoL directly; (2) bullying victimization would be negatively associated with QoL through the mediation of emotional problems; (3) bullying victimization would be negatively associated with QoL through the mediation of ADHD and ODD symptoms; and (4) bullying victimization would be negatively associated with QoL through the sequential mediation of ADHD and ODD symptoms and emotional problems.

## 2. Methods

### 2.1. Participants

This study was conducted in two child psychiatric outpatient clinics in Taiwan between November 2009 and July 2012. The inclusion criteria were adolescents who were 12 to 18 years old and had the diagnosis of ADHD based on the DSM-IV-TR [54]. The exclusion criteria were adolescents who had a diagnosis of intellectual disability, schizophrenia and other psychotic disorders, bipolar disorder, major depressive disorder, or pervasive developmental disorder based on their chart records, as well as those who exhibited difficulty in any cognitive function or verbal communication that prevented them from comprehending the study aims, procedure or questionnaires. Adolescents who met the inclusion criteria were consecutively invited to participate in this study. A child psychiatrist interviewed the adolescents’ caregivers, based on the ADHD module of the Mini-International Neuropsychiatric Interview for Children and Adolescents [55], to confirm the diagnosis of ADHD. A total of 171 adolescents with ADHD (36 girls and 135 boys, mean age = 14.0 years, standard deviation [SD] = 1.5 years) and their caregivers participated in this study. This study was approved by the Institutional Review Board of Kaohsiung Medical University Hospital (approval number: KMUH-IRB-980519; date of approval: 15 April 2010).

### 2.2. Measures

#### 2.2.1. Bullying Victimization

Adolescents’ self-reported victimization of school bullying in the previous 12 months before this study were assessed by using the eight items of victimization subscale on the Chinese version of the School Bullying Experience Questionnaire (SBEQ) in Traditional Chinese [56]. These eight items evaluate adolescents’ experiences of victimization of verbal and relational bullying (4 items, including: being left out during recess or lunch time; being neglected in chats; being called by a mean nickname; being spoken ill of), and physical bullying and the snatching of belongings (4 items, including: being beaten up; being forced to do work; having money taken away; having school supplies or snacks taken away). Each item was rated on a Likert 4-point scale ranging from 0 (never), 1 (just a little), 2 (often), to 3 (all the time). The C-SBEQ has been found to have acceptable one-month test–retest and internal consistency reliability and congruent validity among adolescents in Taiwan [57]. The internal consistency reliability of the victimization subscale on the C-SBEQ in the present study was acceptable (Cronbach’s α coefficient of verbal and relational bullying, and physical bullying and the snatching of belongings were 0.70 and 0.71, respectively).

#### 2.2.2. QoL

Adolescents’ self-reported QoL in the previous 12 months before this study were assessed by using the Taiwanese Quality of Life Questionnaire for Adolescents (TQOLQA) in Traditional Chinese [58]. The 38-item TQOLQA assesses seven domains of QoL, including family, residential environment, personal competence, social relationships, physical appearance, psychological well-being, and pain. Each item was rated on a 5-point Likert scale from 1 (not at all), 2 (not much), 3 (average), 4 (quite a bit), to 5 (very much). The high total scores indicated good QoL. A previous study among adolescents in Taiwan found that the TQOLQA had acceptable reliability and validity [58]. The Cronbach’s α coefficient in the present study was 0.88.

#### 2.2.3. Emotional Problems

The variables of emotional problems in the present study consisted of depression and anxiety. The 27-item Children’s Depression Inventory-Taiwanese Version (CDI-TW) in Traditional Chinese was used to assess adolescents’ self-reported depressive symptoms including negative mood, anhedonia, ineffectiveness, interpersonal problems, and negative self-esteem, in the previous one month before this study [57,59]. For each item, the adolescent was presented with three choices that corresponded to three levels of symptomatology: 0 (absence of symptoms), 1 (mild or probable symptom), or 2 (definite symptom). The CDI-TW had acceptable reliability and validity in Taiwanese children and adolescents [57]. The Cronbach’s α value for the CDI-TW in the present study was 0.83. The 39-item Taiwanese version of the Multidimensional Anxiety Scale for Children (MASC-T) in Traditional Chinese was used to assess adolescents’ self-reported anxiety symptoms including physical symptoms, harm avoidance, social anxiety, and separation/panic, in the previous one month before this study [60,61]. Each item was rated on a 4-point Likert scale ranging from 0 (never true about me), 1 (rarely true about me), 2 (sometimes true about me), to 3 (often true about me). The MASC-T had acceptable reliability and validity in Taiwanese children and adolescents [61]. The Cronbach’s α value for the MASC-T in the present study was 0.88.

#### 2.2.4. ADHD and ODD Symptoms

Caregivers of adolescents with ADHD were invited to completed the 26-item shortened Chinese version of the Swanson, Nolan, and Pelham Version IV Scale (SNAP-IV) [62,63], to assess the severity of ADHD and ODD symptoms in the previous month before this study. Each item was rated on a 4-point Likert scale from 0 (not at all), 1 (just a little), 2 (quite a bite), to 3 (very much). The shortened Chinese version of the SNAP-IV had excellent reliability and validity among children and adolescents in Taiwan [62]. The Cronbach’s α values of the inattention, hyperactivity–impulsivity, and ODD subscales in this study were 0.86, 0.90, and 0.91, respectively.

### 2.3. Procedure and Statistical Analysis

The adolescents completed the C-SBEQ, TQOLQA, CDI-TW, and MASC-T. The caregivers completed the SNAP-IV. Pearson’s correlation was used to examine the correlations between bullying victimization, ADHD and ODD symptoms, emotional problems, and QoL. Structural equation modeling (SEM) was used to test the adequacy of the model using Amos version 18.0 software (SPSS, Chicago, IL, USA) [64]. The maximum likelihood method was used to analyze the data. The goodness of fit index (GFI), comparative fit index (CFI), root mean square error of approximation (RMSEA), and standardized root mean square residual (SRMR) were calculated to evaluate the goodness of fit of the model [65]. According to conventional goodness of fit requirements, the GFI and CFI should be >0.9, and RMSEA should be <0.05. The SRMR values of <0.05 and 0.05–0.09 indicate good and acceptable fit, respectively [65]. A two-tailed *p* value of <0.05 was considered statistically significant.

## 3. Results

### 3.1. Correlation Matrices

Table 1 lists the means, standard deviations, and correlation matrices of the measured variables. The victimization of verbal and relational bullying, the victimization of physical bullying, and the snatching of belongings were positively correlated with depression and anxiety and negatively correlated with QoL. The victimization of verbal and relational bullying, but not that of physical bullying and the snatching of belongings, was positively correlated with inattention and hyperactivity–impulsivity. Inattention and oppositional defiance, but not hyperactivity–impulsivity, were negatively correlated with QoL.

### 3.2. Full Model

Our full model had acceptable fit (GFI = 0.935, CFI = 0.942, SRMR = 0.067, and RMSEA = 0.075; Figure 2). The minimum discrepancy between the ideal and actual models was significant (χ^2^ = 1.954, df = 29, *p* = 0.002) because of the effect of sample size [66]. The path from bullying victimization to ADHD and ODD (*t* = 2.53, *p* = 0.012), the path from bullying victimization to emotional problems (*t* = 4.203, *p* < 0.001), and the path from emotional problems to QoL (*t* = −3.77, *p* < 0.001) reached the significant level, whereas the path from bullying victimization to QoL (*t* = 1.197, *p* = 0.231), the path from ADHD and ODD symptoms to emotional problems (*t* = −1.153, *p* = 0.125), and the path from ADHD and ODD symptoms to QoL (*t* = −1.73, *p* = 0.083) did not reach the significant level. Figure 2 presents all standard coefficients for the paths.

### 3.3. Reduced Model

After nonsignificant paths were deleted, the path from bullying victimization to ADHD and ODD (*t* = 2.43, *p* = 0.02), the path from bullying victimization to emotional problems (*t* = 4.43, *p* < 0.001), and the path from emotional problems to QoL (*t* = −10.05, *p* < 0.001) were statistically significant (Figure 3). The reduced model had good fit (GFI = 0.93, CFI = 0.937, SRMR = 0.074, and RMSEA = 0.075). The severity of bullying victimization was positively associated with the severity of emotional problems. In addition, the severity of emotional problems was negatively associated with the level of QoL. Although the severity of bullying victimization was positively associated with the severity of ADHD and ODD symptoms, ADHD and ODD symptoms were not significantly associated with the level of QoL.

## 4. Discussion

The results of the present study revealed that bullying victimization indirectly correlated with QoL through the mediation of emotional problems in adolescents with ADHD. ADHD and ODD symptoms did not mediate the association between bullying victimization and QoL.

### 4.1. Association of Bullying Victimization with QoL and Mediating Effect of Emotional Problems

Bullying victimization is a distressing experience for adolescents who are at the stage of life characterized by the development of social relationships and self-identity. Moreover, bullying victimization harms not only the physical health but also the mental health of adolescents [67]. Thus, we hypothesized that bullying victimization can worsen victims’ generic QoL directly. However, in the present study, the Pearson’s correlation results indicated that victimization involving verbal or relational bullying, physical bullying, and the snatching of belongings correlated significantly with QoL. In contrast, SEM revealed that bully victimization did not correlate directly with QoL, but indirectly correlated with QoL through the mediation of emotional problems in adolescents with ADHD. Several etiologies may account for the result. Bullying victimization may hurt adolescents’ state of mind and result in various adjustment problems. Emotional problems may occur among adolescents who lack the ability and resources to cope with bullying victimization and related adjustment problems [67,68]. Emotional problems may worsen not only the psychological component state but also other components of QoL. QoL incorporates the persons’ physical health, psychological state, level of independence, social relations, personal beliefs, and relationship to salient features of the environment [13]. Emotional problems may alter the mechanism of biological pathways and neurotransmitters and further increase the tendency of physical discomforts such as pain [69]. Adolescents with emotional problems may experience difficulties in enjoying their social lives and withdrawal from social relationships. Emotional problems may also limit ADHD adolescents’ physical and mental capacity to accomplish their obligations at home, at school, or in their extracurricular activities; adolescents may be vulnerable to others’ negative feedbacks that may compromise adolescents’ self-esteem and beliefs. Alternatively, depression and anxiety may impair the social skills and self-esteem of adolescents and make them more susceptible to being victimized by their peers. Therefore, adolescents with ADHD who experience bullying victimization and have emotional problems may have decreased QoL in multiple domains.

### 4.2. Mediating Effect of ADHD and ODD Symptoms

The SEM findings of this study revealed that bullying victimization was positively associated with ADHD and ODD symptoms among adolescents with ADHD. Previous studies have demonstrated that when victims were exposed to violence during sensitive developmental periods, their developing brain can be affected and that this may lead to neurobiological changes in response to the activation of physiological stress response systems, hence impairing their behavioral regulation [44,70,71,72]. Moreover, previous studies have shown that ADHD symptoms were negatively associated with health-related QoL [45,46], indicating that ADHD symptoms may impair individuals’ school performance, peer relations, and family interaction and therefore reduce their QoL. However, the present study did not identify a significant association of exaggerated ADHD and ODD symptoms with QoL in adolescents with ADHD. Our findings indicated that ADHD and ODD symptoms did not mediate the association between bullying victimization and QoL. This result runs contrary to the result of a previous study on adults with ADHD [42]. Additional studies should examine whether the mediating effect of ADHD and ODD symptoms varies across groups of individuals that differ by age and status of treatment.

### 4.3. Implications

On the basis of the results of this study, we suggest that prevention, early detection, and active intervention for bullying victimization are crucial to ensure better QoL of adolescents with ADHD. Well-designed bullying prevention programs are available such as the Olweus Bullying Prevention Program for the general child and adolescent population [5]. However, bullying prevention programs should be modified according to the special needs of adolescents with ADHD; for example, factors related to bullying victimization among adolescents with ADHD should be considered when developing prevention and intervention programs [73]. As emotional problems mediate the association between bullying victimization and poor QoL, physicians and mental health professionals must routinely assess emotional problems among ADHD adolescents with experience of bullying victimization. Early detection and intervention for emotional programs may reduce the effect of bullying victimization on QoL. Helping individuals with ADHD with bullying victimization develop skills to cope with bullying may also reduce their risk of emotional problems.

### 4.4. Limitations

Some limitations of this study should be considered when interpreting the results of this study. First, the cross-sectional study design made it impossible to determine the temporal relationships among variables. Second, information regarding bullying victimization was collected on the basis of adolescents’ self-reports. Collecting information from multiple informants such as classmates, teachers, and caregivers is needed to confirm the results of this study. Third, there are cultural factors that influence the concepts of bullying victimization and perpetration in children and adolescents of various cultural backgrounds. The terms used for describing bullying-related phenomena, their meaning, and the behavioral manifestations of bullying vary in different countries [74]. Such differences may be due to societal differences in cultural value dimensions such as individualism–collectivism; schooling arrangements such as age bands, lunchtime arrangements, and the extent of homeroom class teaching; and the conceptions of abuse [74]. Whether the results of this study can be replicated in other cultures warrants further study.

## 5. Conclusions

The present study indicated that bullying victimization reduced QoL through the mediator of emotional problems. In addition, the findings indicated that bullying victimization was associated with increased ADHD and ODD symptoms in adolescents with ADHD. We suggest that bullying victimization should be actively prevented, detected early, and intervened on to ensure better QoL in adolescents with ADHD. Moreover, emotional problems should be alleviated among adolescents with ADHD with bullying victimization experience to maintain their QoL.

## Figures and Tables

**Figure 1 ijerph-18-09470-f001:**
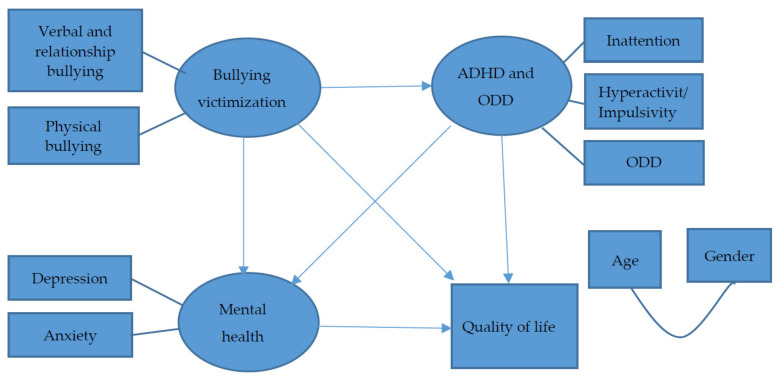
Hypothesized model. ADHD: Attention-deficit/hyperactivity disorder; ODD: Oppositional defiant disorder; Impulsivity: impulsivity subscale of Swanson, Nolan, and Pelham Version IV Scale (SNAP-IV); Inattention: inattention subscale of SNAP-IV; ODD: ODD subscale of SNAP-IV; Depression: total scores of Children’s Depression Inventory-Taiwanese Version; Anxiety: total scores of Taiwanese version of the Multidimensional Anxiety Scale for Children; Verbal and relationship: verbal and relationship subscale of the School Bullying Experience Questionnaire; Physical: physical bullying subscale of the School Bullying Experience Questionnaire.

**Figure 2 ijerph-18-09470-f002:**
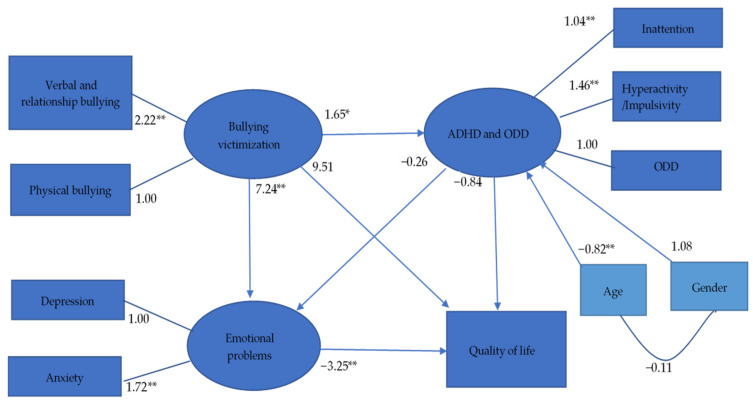
Unstandardized coefficients in hypothesized model. ADHD: Attention-deficit/hyperactivity disorder; ODD: Oppositional defiant disorder. * *p* < 0.05; ** *p* < 0.01.

**Figure 3 ijerph-18-09470-f003:**
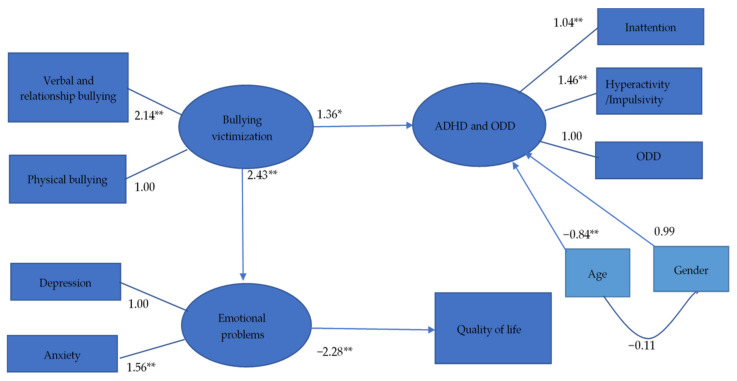
Standard coefficients in reduced model. ADHD: Attention-deficit/hyperactivity disorder; ODD: Oppositional defiant disorder. * *p* < 0.05; ** *p* < 0.01.

**Table 1 ijerph-18-09470-t001:** Correlation matrix of measured variables.

	*n* (%)/Mean (SD)	1	2	3	4	5	6	7	8	9	10
1.	36 (21.1)	-	0.18 *	0.02	0.18 *	−0.05	−0.03	0.01	−0.14	−0.18 *	0.14
2.	14 (1.48)		-	−0.14	−0.32 **	−0.16 *	0.07	0.04	0.06	0.08	−0.10
3.	15.62 (5.70)			-	0.67 **	0.51 **	0.17 *	0.12	−0.05	0.16 *	−0.19 *
4.	9.66 (6.38)				-	0.59 **	0.16 *	0.07	−0.09	0.05	−0.08
5.	11.98 (6.15)					-	0.14	0.07	−0.08	0.17 *	−0.23 **
6.	2.44 (2.50)						-	0.35 **	0.44 **	0.38 **	−0.31 **
7.	0.91 (1.41)							-	0.25 **	0.36 *	−0.29 **
8.	35.59 (18.08)								-	0.54 **	−0.46 **
9.	15.56 (7.64)									-	−0.72 **
10.	131.53 (20.03)		-	-	-	-	-	-	-		-

1: Female gender; 2: Age; 3: Inattention on the subscale of the SNAP-IV; 4: Hyperactive and impulsivity on the subscale of SNAP-IV; 5: Oppositional defiance on the subscale of SNAP-IV; 6: Depression on the Children’s Depression Inventory; 7: Anxiety on the Multidimensional Anxiety Scale for Children; 8: Verbal and relationship bullying victimization on the School Bullying Experience Questionnaire (SBEQ); 9: Physical bullying victimization on the SBEQ; 10: Quality of life on the Taiwanese Quality of Life Questionnaire for Adolescents. * *p* < 0.05; ** *p* < 0.01.

## Data Availability

The data will be available upon reasonable request to the corresponding authors.

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
