# Peer review of "Relationship between Bullying Victimization and Quality of Life in Adolescents with Attention-Deficit/Hyperactivity Disorder (ADHD) in Taiwan: Mediation of the Effects of Emotional Problems and ADHD and Oppositional Defiant Symptoms"

_ijerph, 2021, doi:10.3390/ijerph18189470_

Round 1
Reviewer 1 Report
This article makes a useful contribution, but will benefit from some revision.
Introduction
Line 50 the ‘previous study’ was in Turkey and that should be indicated.
In fact, throughout, it would be useful to be told the country in which studies were carried out in. This study is in Taiwan, but most previous studies were probably in western countries. There are important cultural factors at play in bullying and victimization, and this is the issue that needs most attention in the Introduction and in the Discussion.
It also emerges in Methods! In section 2.2.1, we should be told what language was used in the questionnaires. It is particularly important to say if the word ‘bully’ (or ‘bullying’) was used anywhere and if so how it was translated. Of the 8 victimization items, we are told the wording of 6 of them – please add the other 2.
The Results and Discussion are well done, and the main limitations mentioned. Generally, the paper is clearly written and concise, and a pleasure to read. However greater sensitivity to cultural context throughout would considerably enhance it.
Author Response
We appreciated your valuable comments. As discussed below, we have revised our manuscript based on your comments. Please let us know if we need to provide anything else regarding this revision.
Comment 1
Line 50 the ‘previous study’ was in Turkey and that should be indicated. In fact, throughout, it would be useful to be told the country in which studies were carried out in. This study is in Taiwan, but most previous studies were probably in western countries. There are important cultural factors at play in bullying and victimization, and this is the issue that needs most attention in the Introduction and in the Discussion.
Response
Thank you for your suggestion.
- We added the countries in which the studies of reference 12 (Turkey), 15 (USA) and 18 (Spain) on bullying were carried out into the revised manuscript. Please refer to Line 51, 63 and 78.
- We agree that the studies on bullying and victimization warrants taking cultural differences into consideration. In Discussion section, we added a paragraph as below to emphasize its importance. Please refer to Line 340-347.
“There are cultural factors that influence the concepts of bullying victimization and perpetration in children and adolescents of various cultural backgrounds. The terms used for describing bullying-related phenomena, their meaning, and the behavioral manifestations of bullying vary in different countries [47]. Such differences may be due to societal differences in cultural value dimension dimensions such as individualism-collectivism; schooling arrangements such as age bands, lunchtime arrangements, extent of home-room class teaching; and the conceptions of abuse [47]. Whether the results of this study can be replicated in other cultures warrants further study.”
Comment 2
It also emerges in Methods! In section 2.2.1, we should be told what language was used in the questionnaires. It is particularly important to say if the word ‘bully’ (or ‘bullying’) was used anywhere and if so how it was translated. Of the 8 victimization items, we are told the wording of 6 of them – please add the other 2.
Response
- Traditional Chinese was used in the questionnaires. We added it into the introduction of the questionnaires. Please refer to Line 148-149, 164, 174 and 182.
- The term “bullying” is translated into “bàling” in Taiwan. However, the Chinese version of the School Bullying Experience Questionnaire used in this study describes the bullying behaviors but not the term “bullying” directly.
- We described all 8 items of victimization as below into section 2.2.1. Please refer to Line 150-154.
“…being left out during recess or lunch time; being neglected in chats; being called by a mean nickname; being spoken ill of; being beaten up; being forced to do work; having money taken away; having school supplies or snacks taken away…”
Comment 3
The Results and Discussion are well done, and the main limitations mentioned. Generally, the paper is clearly written and concise, and a pleasure to read. However greater sensitivity to cultural context throughout would considerably enhance it.
Response
Thank you for your comment. As the response to Comment 1, we added a paragraph to emphasize the importance of cultural contexts.
Reviewer 2 Report
This significant paper focuses on the mediating effects of emotional problems, ADHD, and ODD symptoms in the association between bullying victimization and QOL. Accumulating such research will help identify the difficulties of adolescents with ADHD and/or ODD symptoms and lead to the development of more appropriate support methods. However, I have some concerns, as mentions below.
< Major comments >
- ADHD, which is characterized by high impulsivity, is associated with disruptive behavior disorder such as ODD and CD, so I think that the symptoms of ODD were also measured. However, there was little explanation as to why ODD is treated in this study. Please describe a more specific description of the relationship between ADHD and ODD.
- Regarding the hypothesized model, as described in Line 45-48, considering that ADHD and ODD symptoms may increase bullying victimization, I think it would be better to draw a pass from ADHD and ODD symptoms to bullying victimization instead of drawing a pass from bullying victimization to ADHD and ODD symptoms. Furthermore, I think that the emotional problems and stress responses that result from bullying victimization exacerbate ADHD and ODD symptoms, rather than bullying victimization itself exacerbating ADHD and ODD symptoms. In this case, I think it would be better to draw a pass from emotional problems to ADHD and ODD symptoms instead of drawing a pass ADHD and ODD symptoms to emotional problems. Please reexamine the hypothesized model.
- Since ADHD and ODD are strongly related to sex and age, it is better to include sex and age as covariates in the analysis model. In addition, I think it is better to attach observation variables (subscales of each latent variable) to the latent variables and show them in the figure.
I hope that the above comments and suggestions may be helpful for you.
Author Response
We appreciated your valuable comments. As discussed below, we have revised our manuscript based on your comments. Please let us know if we need to provide anything else regarding this revision.
Comment 1
ADHD, which is characterized by high impulsivity, is associated with disruptive behavior disorder such as ODD and CD, so I think that the symptoms of ODD were also measured. However, there was little explanation as to why ODD is treated in this study. Please describe a more specific description of the relationship between ADHD and ODD.
Response
Thank you for your comment. We added more explanation for assessing the role of ODD in this study as below. Please refer to Line 82-88.
“Oppositional defiant disorder (ODD) is comorbid in more than half of individuals with ADHD [21]. Comorbid symptoms of ODD in individuals with ADHD can have a significant impact on the course and prognosis and may lead to differential treatment response to both behavioral and pharmacologic treatments. Therefore, ODD symptoms are a key consideration in ADHD management [21]; the comorbidity of ODD exerted an additionally deleterious effect on the QoL of individuals with ADHD [22].”
Comment 2
Regarding the hypothesized model, as described in Line 45-48, considering that ADHD and ODD symptoms may increase bullying victimization, I think it would be better to draw a pass from ADHD and ODD symptoms to bullying victimization instead of drawing a pass from bullying victimization to ADHD and ODD symptoms. Furthermore, I think that the emotional problems and stress responses that result from bullying victimization exacerbate ADHD and ODD symptoms, rather than bullying victimization itself exacerbating ADHD and ODD symptoms. In this case, I think it would be better to draw a pass from emotional problems to ADHD and ODD symptoms instead of drawing a pass ADHD and ODD symptoms to emotional problems. Please reexamine the hypothesized model.
Response
Thank you for your suggestion. We reexamined the hypothesized model based on your suggestion, including draw a pass from ADHD and ODD symptoms to bullying victimization as well as the paths from bullying victimization to emotional problems and then to ADHD and ODD symptoms. However, the fit of new SEM was poor (GFI = 0.763, CFI = 0.762, SRMR = 0.0564, and RMSEA = 0.133). The result did not support the fitness of the new model.
Comment 3
Since ADHD and ODD are strongly related to sex and age, it is better to include sex and age as covariates in the analysis model. In addition, I think it is better to attach observation variables (subscales of each latent variable) to the latent variables and show them in the figure.
Response
We have included sex and age as covariates in the original analysis model. We added the explanation into the revised manuscript. Please refer to Line 101. We also showed the subscales of each latent variable in Figure 1. Please refer to Line 108-117.
Round 2
Reviewer 2 Report
Thank you for revising the manuscript. I have some concerns below.
In Table 1, please show the correlation coefficient between the covariates (age and gender) and each variable.
Is the path to each variable from age and gender omitted in Figure 1?
Also, in Figure 1, is age and gender connected by a two-way path?
Path lengths and angles are inconsistent in all figures, so please align them neatly for readability.
In all figures, please do not delete the observed variables connected to the latent variables. Also, please show each path coefficient in the observed variable.
In all figures, "emotional problems" is "mental health". Please unify the notation name to either "emotional problems" or "mental health".
Author Response
We appreciated your valuable comments. As discussed below, we have revised our manuscript based on your comments. Please let us know if we need to provide anything else regarding this revision.
Comment 1
In Table 1, please show the correlation coefficient between the covariates (age and gender) and each variable.
Answer
Thanks for your suggestions. We added the correlation coefficients into Table 1.
Comment 2
Is the path to each variable from age and gender omitted in Figure 1?
Answer
Based on your suggestions, we controlled the effects of gender and age on ADHD and ODD. We added the paths from gender and age to ADHD and ODD into three figures.
Comment 3
In Figure 1, is age and gender connected by a two-way path?
Answer
We argued that age and gender showed a high relation to each other; therefore, we applied a two-way path between them.
Comment 4
Path lengths and angles are inconsistent in all figures, so please align them neatly for readability.
Answer
Thanks for your suggestions. We revised the figures in the revised manuscript.
Comment 5
In all figures, please do not delete the observed variables connected to the latent variables. Please show each path coefficient in the observed variable.
Answer
Thanks for your suggestion. We added the observed variables connected to the latent variables into the figures in the revised manuscript. We also showed the standardized coefficient in the observed variables in the revised manuscript.
Comment 6
In all figures, "emotional problems" is "mental health". Please unify the notation name to either "emotional problems" or "mental health"
Answer
Thank you for your comment. We unified the term into "emotional problems" in all figures.